# Endovascular progenitors infiltrate melanomas and differentiate towards a variety of vascular beds promoting tumor metastasis

Prudence Donovan[1], Jatin Patel [1], James Dight[1], Ho Yi Wong[1], Seen-Ling Sim[1], Valentine Murigneux[1], Mathias Francois[2] & Kiarash Khosrotehrani[1]

Tumor vascularization is a hallmark of cancer central to disease progression and metastasis. Current anti-angiogenic therapies have limited success prompting the need to better understand the cellular origin of tumor vessels. Using fate-mapping analysis of endothelial cell populations in melanoma, we report the very early infiltration of endovascular progenitors (EVP) in growing tumors. These cells harbored self-renewal and reactivated the expression of SOX18 transcription factor, initiating a vasculogenic process as single cells, progressing towards a transit amplifying stage and ultimately differentiating into more mature endothelial phenotypes that comprised arterial, venous and lymphatic subtypes within the core of the tumor. Molecular profiling by RNA sequencing of purified endothelial fractions characterized EVPs as quiescent progenitors remodeling the extracellular matrix with significant paracrine activity promoting growth. Functionally, EVPs did not rely on VEGF-A signaling whereas endothelial-specific loss of *Rbpj* depleted the population and strongly inhibited metastasis. The understanding of endothelial heterogeneity opens new avenues for more effective anti-vascular therapies in cancer.

[1] Translational Research Institute, UQ Diamantina Institute, The University of Queensland, Brisbane 4102 QLD, Australia. [2] Institute of Molecular Biosciences, The University of Queensland, Brisbane 4072 QLD, Australia. These authors contributed equally: Prudence Donovan, Jatin Patel. Correspondence and requests for materials should be addressed to J.P. (email: j.patel@uq.edu.au) or to K.K. (email: k.khosrotehrani@uq.edu.au)

Tumor vascularization is one of the hallmarks of cancer. It is classically proposed that for tumor progression to occur new blood vessels must form that will allow the provision of oxygen and nutrients, however these vessels also provide beneficial roles by allowing immune cells and drug delivery to inhibit tumor growth[1]. Furthermore, blood vessels have angiocrine capacity supporting directly the growth of tumors through the production of cytokines and growth factors[2]. They have also been proposed to facilitate tumor spread through the blood or lymphatic vasculature[3]. More recently, it has also been proposed that tumor vessels behaving abnormally contribute to the hypoxic environment and hence maintain tumor cells in an invasive state[4]. Beyond its detrimental role, tumor vasculature is an essential component of drug and immune cell delivery to the tumor. Overall these properties have prompted numerous attempts at normalizing abnormal blood vessel formation in the context of cancer rather than outright abrogating tumor vessels[5].

Tumors are vascularized through a variety of modalities but rely predominantly on angiogenesis where VEGF family members play a crucial role. However, anti-VEGF therapy has failed in many indications to reduce tumor size, spread, or vascularization[5]. Although many of the molecular factors that drive tumor vascularization are well known and established, much less is known about the cellular origin of vessel network in tumors. In particular, it is often unclear which vascular bed or which cells are at the source of newly formed vessels in primary tumors. Past studies have proposed the existence of endothelial progenitors driving tumor vascularization[6]. Contrasting early studies, the hematopoietic, bone marrow derived, and circulating nature of this endothelial progenitors has been refuted[7] and it has been clarified that these cells are of a resident endothelial nature[8]. Although informative, many of these studies relied on single markers or cell transfers as opposed to cell fate mapping of endogenous progenitors as well as a functional rather than a marker-based definition of progenitors[9,10].

We have recently reported in a variety of vascular beds in mice[11] and humans[12] that the adult endothelium is heterogeneous and is composed of 3 distinct populations: an endovascular progenitor (EVP), a transit amplifying (TA), and a differentiated (D) population. In the present study, we have extensively examined the formation of tumor vasculature and show that very early upon inoculation, tumors are infiltrated by SOX18 expressing EVP cells that originate from arterial or venous but not from lymphatic beds. These EVP cells give rise to TA and D cells that form mostly venous/arterial capillaries but also lymphatics through the combined contribution of multiple clones of endothelial cells. At the functional level, only EVP cells have colony forming and transplantation capacity. The molecular characterization of EVPs shows significant differences with TA and D cells. Critically for a therapeutic perspective, anti-VEGF-A therapy did not affect EVP cells. On the other hand, conditional ablation of RBPJ, a direct protein interactor of SOX18[13,14], dramatically reduced EVP cells and resulted in the abrogation of metastases providing perspectives for anti-vascular therapy of cancer by targeting the EVP population.

## Results

### Functional and molecular heterogeneity of tumor vasculature.
To explore the cellular origin of vessels in a growing tumor and establish the kinetics of vessel assembly, we undertook orthotopic delivery of B16-F0 melanoma cells intradermally. *Cdh5-Cre^ER RosaYFP* mice were used to label all endothelial cells with YFP using tamoxifen injection for 5 consecutive days. Subsequently, mice were inoculated intradermally with B16-F0 tumor cells. Upon injection and development, tumors were visible from

5–7 days macroscopically and could be easily distinguished from surrounding tissues. We first examined the heterogeneity of endothelial cells based on variation in cell surface markers[11]. Dissection, single cell suspension, and analysis of tumors with minimum contamination by surrounding cells using multi-color flow cytometry allowed determining positive and negative staining for every marker using the fluorescence minus one (FMO) method (Supplementary Figure 1A). In the absence of tamoxifen injection no YFP could be observed (not shown). Using a live gate, doublets were discounted and hematopoietic cells were identified using the lineage cocktail (Lin). Among Lin-CD34+ cells, most cells (>98%) were YFP+ demonstrating their endothelial origin (Fig. 1a). Among these VE-cadherin expressing cells, three populations could be easily identified based on their levels of CD31 and VEGFR2 identified as EVP (CD31^lo VEGFR2^lo), TA (CD31^int, VEGFR2^lo), and D (CD31^hi VEGFR2^hi)[11]. We also ensured that these Lin-CD34+ populations did not express the hematopoietic progenitor marker c-kit (Supplementary Figure 1B). We further ensured the validity of our analysis using spontaneous melanomas developed on *Tyr::Nras; Cdk4r24c*[15] transgenic mice as well as in Lewis Lung Cancer (LLC) and EO771 breast cancer tumors and observed similar endothelial heterogeneity (Supplementary Figure 1C & D).

We next evaluated the functional differences between the three populations. Here, we injected B16-F0 cells in CAG-GFP transgenic mice ubiquitously expressing GFP (Fig. 1b). After 15 days, tumors were removed and GFP+ EVP, TA, and D populations were flow-sorted, mixed with B16-F0 tumor cells, and re-implanted in secondary non-transgenic recipients. After 14 days, secondary tumors were removed, weighed, and subjected to flow cytometry to identify GFP+ endothelial cells. Only EVP cells had a capacity to survive and repopulate, whereas TA and D cells inoculated in secondary tumors could not be recovered 14 days later (**$p < 0.01$ vs TA and D) (Fig. 1c, d) and visualized in tumor sections (Fig. 1e). These cells maintained their endothelial characteristics as they expressed CD34 and were devoid of any hematopoietic lineage marker. Furthermore, delivery of B16-F0 cells mixed with purified EVP resulted in significantly larger tumors (increase in weight by 34% compared to B16-F0 alone) collected 7 days after transplant, whereas tumor cells delivered with D cells alone resulted in smaller tumors (decrease in weight by 33% compared to B16-F0 alone, *$p < 0.05$, Mann–Whitney *T*-Test) (Fig. 1f). Overall, these findings confirm that vascular endothelial cells are heterogeneous in their cell surface molecular markers as well as in their self-renewal and engraftment potential.

### Lineage relationship between endothelial populations.
We next asked whether these three populations derived from one another and used fate mapping analysis to establish their lineage relationship over time. *Cdh5-Cre^ER RosaYFP* mice were inoculated with B16-F0 melanoma cells at D0 and injected with a single dose of tamoxifen on D3 (Fig. 2a). Tumors were collected on subsequent days to establish the fate of endothelial cells labeled. On D5, flow cytometry allowed to clearly identify YFP+ CD34+ cells that did not express any hematopoietic markers (lineage negative) within tumors. Most of these cells were CD31^lo VEGFR2^lo and corresponded to EVP and a few TA cells. At this early time point, no D cell could be identified (Fig. 2a) and tumors sections revealed that YFP+ EVP cells were mostly single cells that only expressed CD34 but not CD31 or VEGFR2 (Fig. 2b, Supplementary Figure 2A). At D10, however, many cells were identified as D with some TA cells. At this time point, no obvious YFP positive EVP cell could be found suggesting that the progenitor pool labeled at D3 had undergone differentiation within the

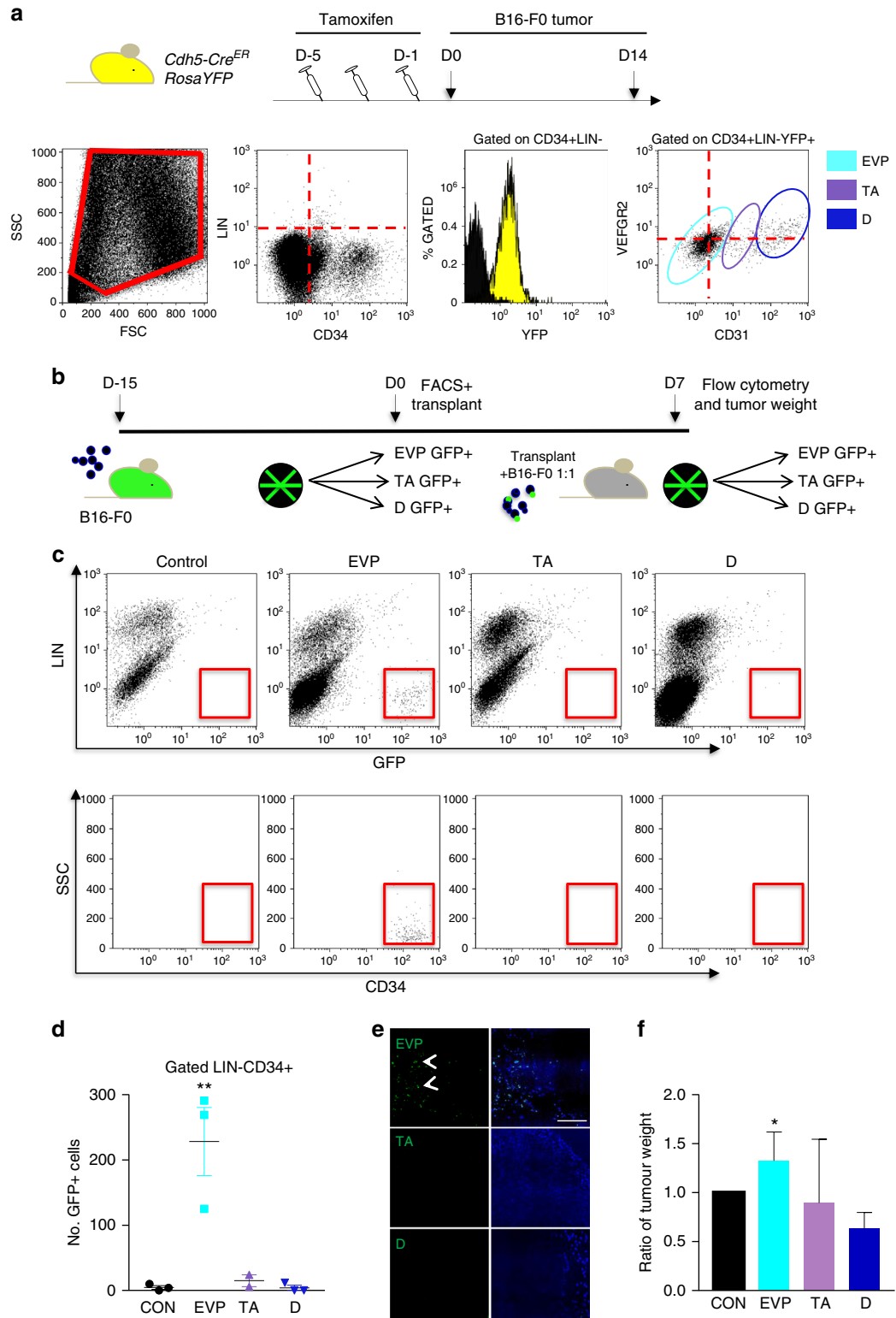

tumors. Finally by D15, D cells were the only population in abundance that was remaining from the cells that were stained on D3, with only small numbers of EVP and TA observed. At D10 and D15, YFP+ cells were organized in vascular structures infiltrating the tumors (Fig. 2b). Overall these findings clarify the transition from EVP to TA to D in a 7–12 days period and demonstrate the lineage relationship between the three populations.

We next examined if the EVP were infiltrating the tumor and most importantly if they were originating from a non-endothelial origin. We therefore used pulse chase of tamoxifen studies in the *Cdh5-Cre^ER RosaYFP* mice over an extended period of time as previously reported[16] (Supplementary Figure 2B). Tamoxifen was initially provided for 5 consecutive days and the normal back skin was processed to assess the endothelial compartment. Greater than 99% of all Lin-CD34+ cells were YFP+, demonstrating the

**Fig. 1** Melanoma endothelium in heterogeneous. **a** Schematic diagram demonstrating experimental set up using vascular lineage tracing *Cdh5-Cre*[ER] *RosaYFP* mice. Flow cytometry plots showing cells dissociated from B16-F0 tumors harbor distinct CD34 positive, lineage (LIN) negative populations (red gate) as determined using strict fluorescence-minus-one (FMO) analysis. >98% of CD34+LIN- cells are YFP+. Three distinct populations were observed based on CD31 and VEGFR2 expression in tumors (from left to right: EVP, TA, and D) amongst CD34+LIN-YFP+cells (*n* = 4). **b** Schematic diagram demonstrating the isolation of GFP+ EVP, TA, and D from tumors, which were subsequently re-transplanted in a 1:1 ratio with B16-F0 cells into a wild-type host. **c, d** Flow cytometry plots showing only GFP+ EVP cells re-transplanted were able to persist and engraft in secondary tumors. TA and D cells inoculated in secondary tumors could not be recovered 14 days later (**\*\***p < 0.01 vs TA and D; Mann–Whitney *T*-Test) (*n* = 6). **e** Immunofluorescence images of only GFP+ EVP cells (white arrows) engrafting and surviving transplantation with B16-F0 (scale bar represents 100 μm). **f** Delivery of B16-F0 cells with GFP+ EVP resulted in larger tumors (increase in weight by 34% compared to B16-F0 alone), whereas tumor cells delivered with D cells alone resulted in smaller tumors (decrease in weight by 33% compared to B16-F0 alone) (*\*p* < 0.05; Mann–Whitney *T*-Test). Results presented as mean ± SEM. EVP endovascular progenitor, TA transit amplifying, D definitive differentiated, CON control (no GFP cells)

entire labeling of the vasculature (Supplementary Figure 2C). Of interest, the normal skin vasculature was also organized in EVP, TA, and D compartments. Next, after a similar pulse of tamoxifen, mice were left for a latency period of 14 days before being inoculated subcutaneously with B16-F0 cells. The tumors were then collected 10 days later, or 24 days after their final tamoxifen injection. Within the tumors greater than 99% of all Lin-CD34+ cells remained YFP+, clarifying that the endothelial population was only originating from the vasculature and not being diluted by other unlabeled populations (Supplementary Figure 2D). Given the rapid development of these tumors that on D5 are devoid of any visible blood vessels, our findings suggest EVPs gain the center of the tumor through infiltration from surrounding vessels.

**Sox18 re-expression EVP progenitors in tumors**. In our previous work, we have shown the critical importance of *Sox18* re-expression in EVPs[11]. Indeed, *Sox18* is a gene involved in embryonic vascular development. Its expression in the vasculature is lost at adult age in endothelial cells except in situations of new vessel formation such as wounds[17] or tumors[18]. We performed similar lineage tracing using *Sox18-Cre ROSA-YFP* mice (Fig. 2c). When induced with tamoxifen at D3 post tumors inoculation, the first YFP+ population identified at D5 was the EVP. At this time point we observed very few TA, no D cells, and EVP cells once again in this different model consisted in single cells (Fig. 2c, d). D populations could be observed at D10–D15 post-tumor inoculation. This clearly implies that EVPs initially express *Sox18* and then give rise to TA and D cells through complete differentiation. This additional model clearly reinforces the observation of transitions between EVP, TA, and D.

We next asked whether the initial input of EVPs was sufficient to drive tumor vascularization or alternatively whether additional EVPs would enter the same process of differentiation. We therefore used *Sox18-Cre ROSA-YFP* mice inoculated with B16-F0 tumors at D0 and induced them with tamoxifen at D3 but also at D8 and collected tumors at D12 (Supplementary Figure 3). The additional stimulation of *Sox18*-induced Cre activity at D8, demonstrated the staining of additional EVP cells that could not be observed by a single injection on D3 alone. This result suggests that there is an ongoing process by which EVP cells arise to sustain a continuous makeup of the tumor vasculature.

**EVPs contribute to both arterial and venous vascular beds**. Having established that in tumors only EVPs have self-renewal potential and differentiate into TA and D cells, we next asked the capacity of these cells to give rise to all possible vascular beds. On D5, EVP cells labeled with YFP were essentially single cells infiltrating the bulk of tumor cells. This was observed for both *Sox18* and *Cdh5*-driven YFP labeling (Fig. 3). They did not express any specific vascular bed marker. By D10 and D15, YFP

cells had formed entire vessel network. The majority of vessels expressed the venous capillary marker endomucin, and about a third of YFP+ vessels displayed arterial markers such as DLL4 or Sox17 (not shown). These observations were made in both models of lineage tracing showing the potential of *Cdh5* or *Sox18* expressing EVPs to give rise to these structures. Overall, the quantification of lineage tracing experiments shows that *Sox18* and *Cdh5* expressing cells seem both to give rise to the same proportions of capillary, veins, and arteries suggesting that all EVPs activate the expression of *Sox18* prior to vessel formation.

**EVPs contribute to lymphatic vascular beds**. Tumor-induced neo-lymphangiogenesis has long been thought to exclusively derive from pre-existing lymphatic vessels[3] or from myeloid cells infiltrating the tumor vascular[19]. In both *Cdh5* and *Sox18*-driven lineage tracing models, a discrete population of YFP+ lymphatic vessels could be observed by D15 within the center of the tumor (Fig. 4a). This was observed using both Lyve-1 and Podoplanin lymphatic markers. The rarity of this event upon quantification reaching upon 6–8% of YFP+ vessels (Fig. 4b) prompted us to examine whether lymphatics had a separate progenitor. We performed lineage tracing experiments using *Prox1Cre tdtomato* (Fig. 4c). On tumor sections obtained on D5 post tumor inoculation (2 days after tamoxifen delivery), the large majority of Prox1+ cells co-localized with Lyve1 and Podoplanin expressing lymphatics in the periphery of the tumor (Fig. 4d). Flow cytometry on D5 showed that *Prox1* (tdTomato) expressing cells were not Lin-CD34+ suggesting that EVPs within tumors at this time point were not derived from the Prox1 expressing lymphatic vascular bed (Fig. 4e). This further suggests that EVPs contribute to lymphatics through the expression of *Sox18* likely similar to the series of embryonic events that where this transcription factor induce lymphatic endothelial cell fate in venous progenitor cells[20]. However, this remains a small fraction of tumor lymphatics. Overall, these findings show that *Sox18* expressing EVPs derive from venous or arterial but not lymphatic beds and contribute to most of the venous capillaries and some of the arterial and lymphatic vessels within tumors.

**EVP from multi-clonal populations derive tumor vessels**. We next wondered how single EVP cells observed on D5 could give rise to entire vessels of arterial or venous nature. In particular, we asked if this was related to a single clone growing into a given vessel or if multiple EVPs would compose a single vessel. We therefore performed multicolor lineage tracing using a *Cdh5-Cre*[ER] *Rainbow* reporter system as previously reported by us[21,22]. In this model, each cell upon *Cre* recombinase activation undergoes a random recombination of one or two Rainbow 1.0 cassette (Fig. 5a) allowing the random expression of one out of 5 different colors. *Cdh5-Cre*[ER] *Rainbow* mice were injected with a low dose (0.3 mg) of tamoxifen at D3 post B16-F0 tumor

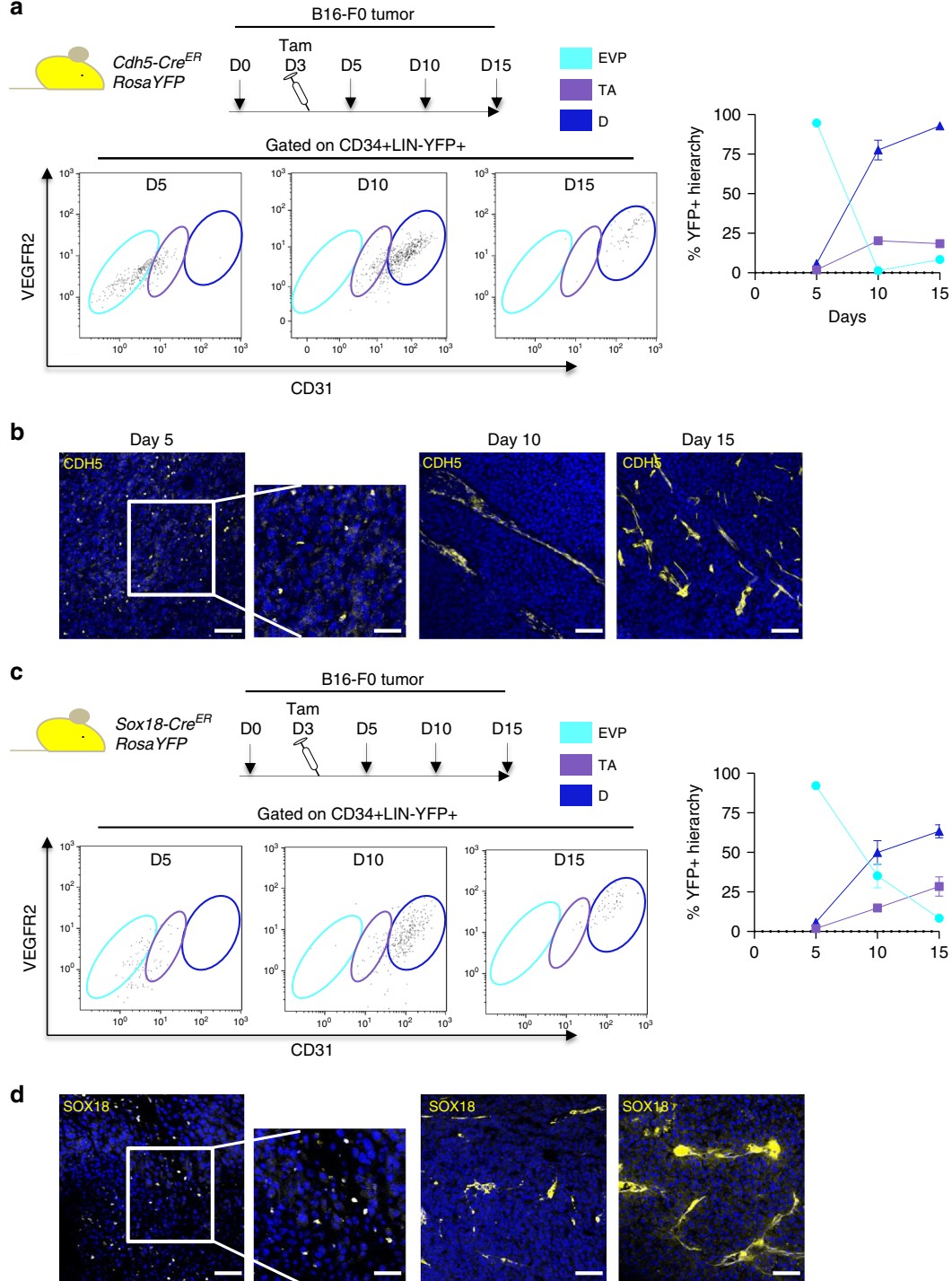

**Fig. 2** Lineage tracing reveals the kinetics of the endothelial hierarchy in tumors. **a** Schematic diagram of the *Cdh5-Cre^ER RosaYFP* lineage tracing model employed with B16-F0 melanoma cells injected at day 0 (D0), with mice receiving tamoxifen (Tam) 3-days post tumor inoculation. Flow cytometry plots demonstrate that D5 post tumor inoculation, only EVP and TA amongst CD34+ LIN-YFP+ cells can be observed. At D10 EVP are absent and by D15 only mature D cells can be observed (*n* = 5). **b** Immunofluorescence at D5 shows individual YFP+ foci. At D10 and 15 entire YFP+ vessel structures can be observed. **c** Schematic diagram of the *Sox18-Cre^ER RosaYFP* lineage tracing model employed with B16-F0 melanoma cells injected at day 0 (D0), with mice receiving tamoxifen (Tam) 3-days post tumor inoculation. Flow cytometry plots demonstrate that D5 post tumor inoculation, only EVP and TA amongst CD34+ LIN-YFP+ cells can be observed. At D10 EVP are absent and by D15 only mature D cells can be observed (*n* = 5). **d** Immunofluorescence at D5 shows individual YFP+ foci. At D10 and 15 entire YFP+ vessel structures can be observed. Scale bar represents 50 and 150 μm, respectively. Results presented as mean ± SEM. EVP endovascular progenitor, TA transit amplifying, D definitive differentiated

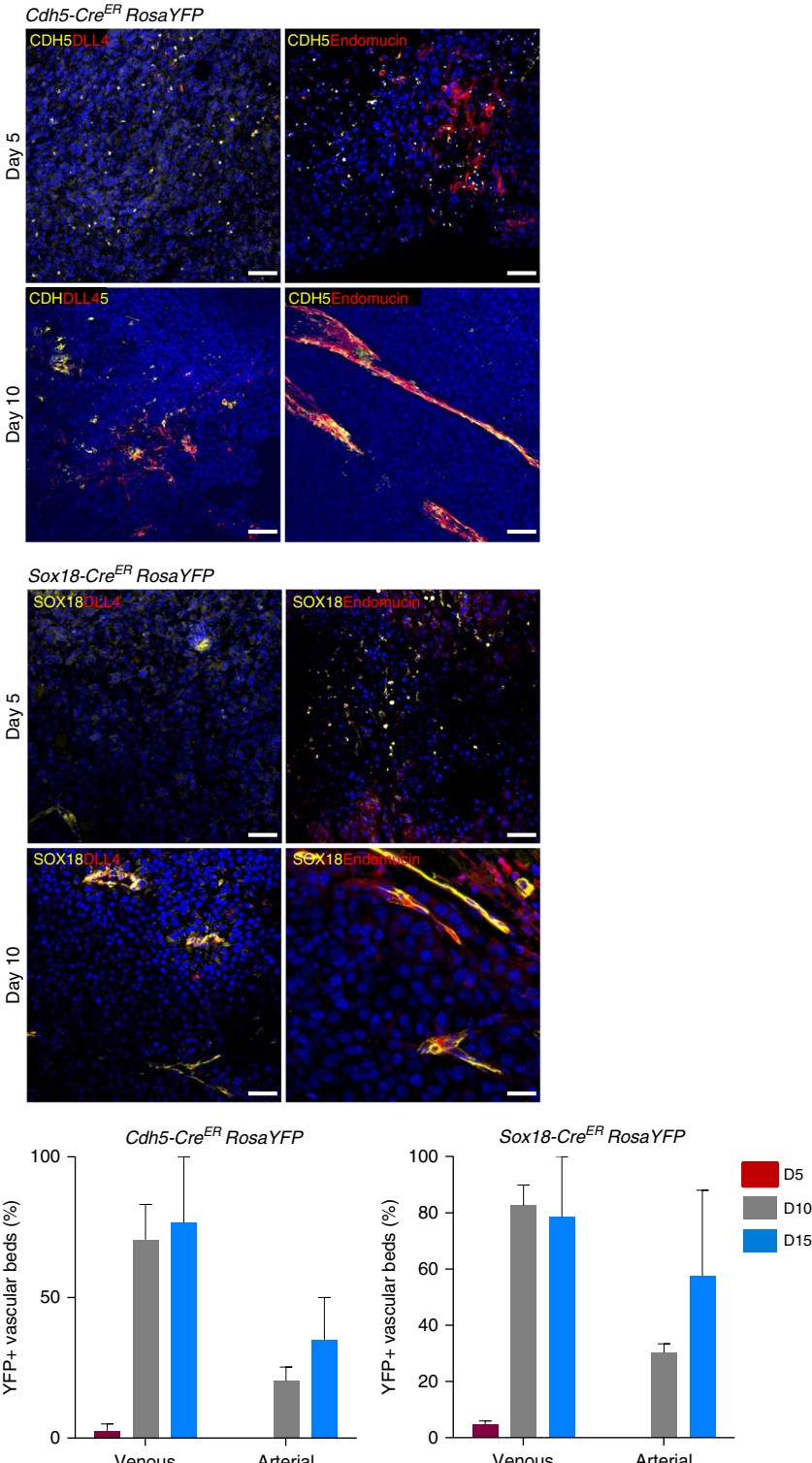

**Fig. 3** EVP contribution to arterial and venous beds. Representative micrographs of tumor sections taken from *Cdh5-Cre^ER RosaYFP* and *Sox18-Cre^ER RosaYFP* lineage tracing models. At day 5 YFP+ cells did not display any arterial or venous markers. By day 10 and 15 YFP+ cells could be co-localized with arterial marker DLL4 and venous marker endomucin ($n = 5$). Results presented as mean ± SEM. Scale bar represents 150 μm

inoculation. Five days post tumor inoculation, multiple EVP clones could be identified within the tumor (Fig. 5b). There was no specific clustering of colors in EVP cells within each tumor suggesting probably the migration of multiple cells stained at D3 rather than the proliferation of a single clone. By D10, both

venous capillaries and arterial vessels seemed composed of more than one color clearly suggesting the contribution of multiple clones to different sections of each vessel within the tumor. However, within each section of vessels labeled with the same color and deemed to be clonal, multiple cells could be identified.

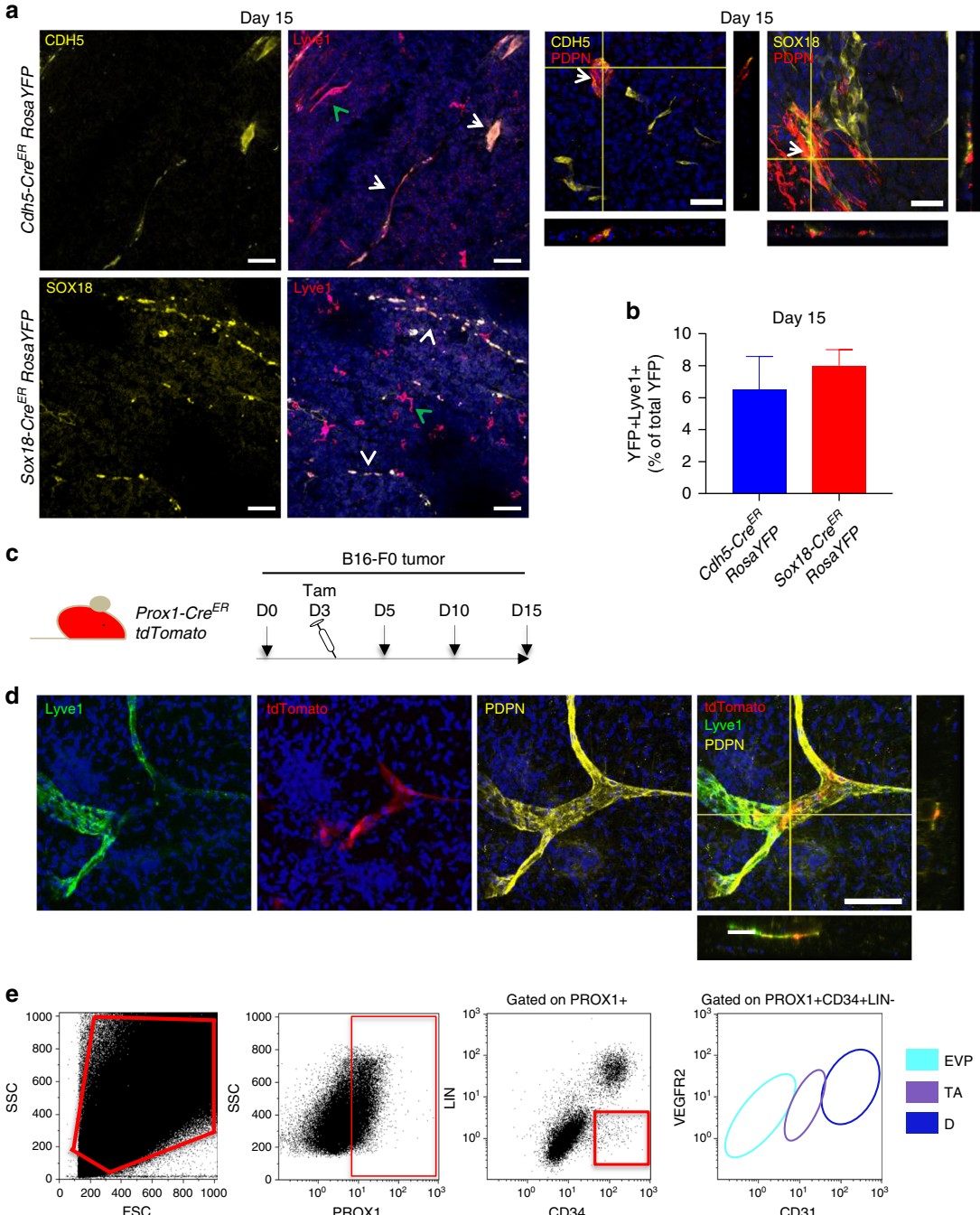

**Fig. 4** EVPs contribute to lymphatic vascular beds but do not originate from lymphatics. **a**, **b** Representative micrographs of tumor sections taken from *Cdh5-Cre^{ER} RosaYFP* and *Sox18-Cre^{ER} RosaYFP* lineage tracing models. At day 15 YFP+ vessels co-localized with lymphatic marker Lyve1 (white arrow) and Podoplanin (PDPN). Green arrows represent Lyve1+ vessels that are not YFP+ (*n* = 5). **c** Schematic diagram of the *Prox1-Cre^{ER} tdTomato* lineage tracing model employed with B16-F0 melanoma cells injected at day 0 (D0), with mice receiving tamoxifen (Tam) 3-days post tumor inoculation (*n* = 5). **d** Lyve1+PDPN+ vessels co-localized with Tomato+ (Prox1) vessels. **e** Flow cytometry plots demonstrating that Prox1+CD34+LIN- cells do not contribute to the endothelial hierarchy. Results presented as mean ± SEM. Scale bar represents 150 μm. EVP endovascular progenitor, TA transit amplifying, D definitive differentiated

This suggests that EVP cells enter a differentiation process as well as some proliferation before joining within the resulting clone to form a vascular structure.

**Molecular characterization of endothelial populations**. Having established the functional hierarchy between EVP, TA, and D endothelial populations and their contribution to venous and arterial beds within the tumor at clonal level, we next aimed to

characterize their molecular profiles. CAG-GFP transgenic mice were injected with B16-F0 tumors. At D15 tumors were collected, GFP+ EVP, TA, and D cells sorted and subjected to RNA sequencing (*n* = 5 per population). The three populations could be separated on PC plots based on their gene expression levels (Fig. 6a). Unsupervised clustering clearly distinguished two groups: node 1 consisted in D cells only whereas node 2 gathered EVP and TA. This clearly shows that the TA cells are more closely related to EVP. Within node 2 however EVP and TA

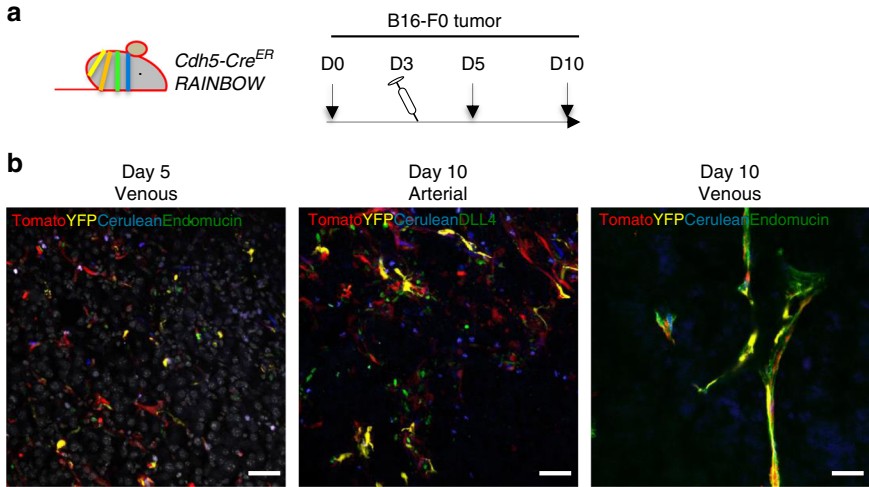

**Fig. 5** Clonality of the tumor vasculature. **a** Schematic diagram of the *Cdh5-Cre^ER Rainbow* lineage tracing model employed with B16-F0 melanoma cells injected at day 0 (D0), with mice receiving tamoxifen (Tam) 3-days post tumor inoculation (*n* = 5). **b** At day 5 multi-individual colored clones could be observed that were presumed to be EVP. By day 10 entire arterial (DLL4) and venous (endomucin-higher magnification) vessels were composed of multi colors demonstrating polyclonality. Scale bar represents 50–150 μm. EVP endovascular progenitors

could be distinguished based on the same clustering (Fig. 6b). Based on this unsupervised clustering, differential gene expression between EVP and D was therefore deemed to be the most important feature. We identified 3064 differentially expressed genes (*p* < 0.05 after multiple testing correction). Reflecting the quality of the cell sorting, *Pecam1* (74×) and K*dr* (VEGFR2, 40×) were largely overexpressed in D compared to EVP as expected (Fig. 6c, Table 1 and Supplementary Figure 4A–F).

**Gene expression reflects state of differentiation**. In general D cells overexpressed many classical endothelial markers such as *Nos3* (enos, 74×), *Vwf* (14×), *Ets1* (9×), *Ets2* (10×), *Gata2* (6×), *Fli1* (22×), *Esam* (47×), *Tie1* (60×), *Cldn5* (111×). As expected from our previous work, D cells also overexpressed genes from the SoxF family (*Sox18*, 52×; *Sox17*, 75×; *Sox7*, 64×). However, as shown above, the expression of *Sox18* despite being higher in D cells, was initiated in EVP cells. In contrast, EVP cells showed some different characteristics. EVP cells displayed mobility with the overexpression of many matrix metalloproteases (MMP; 6 to 293 fold, respectively) and a high capacity to remodel the extracellular matrix (dermatopontin and decorin, >600×; *Col3a1*, 415×; *Col1a2*, 294×; *Has1*, 258×) potentially explaining their ability to infiltrate tumors as single cells. Finally, EVP cells expressed genes classically involved in stem cell function (*Aldh1a1*, 790×; *Sox9* 18×) as well as quiescence (*Nfatc4*, 43×; *Il33*, 409×; *Cdkn1c*, 12×). Looking at significant differences between TA and EVP cells, we identified 580 differentially expressed genes (*p* < 0.05). Once again *Pecam1* and *Kdr* were more highly expressed in TA cells. In many aspects, TA cells had an intermediate gene expression profile between EVP and D cells. Overall, these findings reflect the remarkable heterogeneity within the endothelium and support the hierarchy from progenitor to differentiated cells.

Overall, when performing pathway analysis (DAVID), extracellular matrix–receptor interaction and focal adhesion were the top pathways reflecting the activity of EVP cells. PI3kinase and cytokine receptor signaling were also reflecting some of the key signaling in these cells (Supplementary Figure 4A–F).

**Progenitor cells have significant angiocrine capacity**. EVP cells overexpressed many *Wnt* ligands (*Wnt9a*, 9×; *Wnt5a*, 18×;

*Wnt10b*, 23×; *Wnt16*, 200×; *Wnt11*, 395×; *Wnt2*, 489×), FGFs (*Fgf16*, 8×; *Fgf23*, 8×; *Fgf10*, 9×; *Fgf11*, 11×; *Fgf21*, 11×; *Fgf2*, 13×; *Fgf18*, 76×; *Fgf7*, 165×), Hgf (15×), *Vegfd* (163×), *Vegfa* (9×), *Tgfb3* (14×) and *Pdgfc* (20×). EVPs seemed also able to respond to a range of extracellular growth factors through the expression of *Pdgfra* (205×) and *Pdgfrb* (23×), *Egfr* (106×), *Tgfbr2* (5×), *Tgfbr3* (13×) as well as to a range of cytokines through the expression of *Flt3* (22×), *Il6ra* (13×), *Il11ra1* (11×), and *Il1rl1* (ST2, IL33 receptor; 181×). In particular, cytokine receptor signaling seemed particularly activated given the expression of its downstream targets (*Socs1* and *Socs3*, 5×; *Ifi205*, 305×; *Irf4*, *Irf6* and *Irf7*, 5–30×; *Pim1*, 6×).

**Acquisition of arterial vs venous identity is a late event**. We next wondered whether progenitors were predetermined to give rise to a specific vascular bed. We therefore looked for arterial vs venous or lymphatic markers in EVP, TA, or D cells. Arterial markers such as *Dll4*, *Sox17*, *EphrinB2* (*Efnb2*), *Notch1* and *Notch4*, *Gja5* and *Gja4* were all overexpressed in D cells with little difference in the level of expression between EVP and TA populations suggesting a late differentiation (***p* < 0.001; Mann–Whitney *T*-Test). Regarding venous markers such as COUP-TFII (*Nr2f2*), *Ephb4*, *Nrp2*, or endomucin (*Emcn*), once again the level of expression was higher in D vs EVP suggesting a late differentiation. However, there was also a significant increase in expression in the TA population suggesting the initiation of venous differentiation during the TA stage compared to EVP (**p* < 0.05; ***p* < 0.01; ****p* < 0.001 vs EVP; Mann–Whitney *T*-Test, Supplementary Figure 5A).

**EVPs share common molecular signature in aorta and tumors**. Finally, we have previously compared EVP to D cells in normal aorta[11] and wondered to what extent the tumor and aorta data would overlap. We compared genes differentially expressed between EVP and D in the aorta vs in tumors and found a significant overlap (44% of aorta DEGs and 38% of tumor DEGs, Venn diagram, Supplementary Figure 5B). Genes reflecting stem cell function (*Sox9*, *Aldh1a1*, *Aldh1a2*), quiescence (*Nfatc4*, *Il33*, *Cdn1c*) as well as many of the receptors and paracrine function described above had been also identified in the aorta (Supplementary Figure 5B). We next compared the level of differential

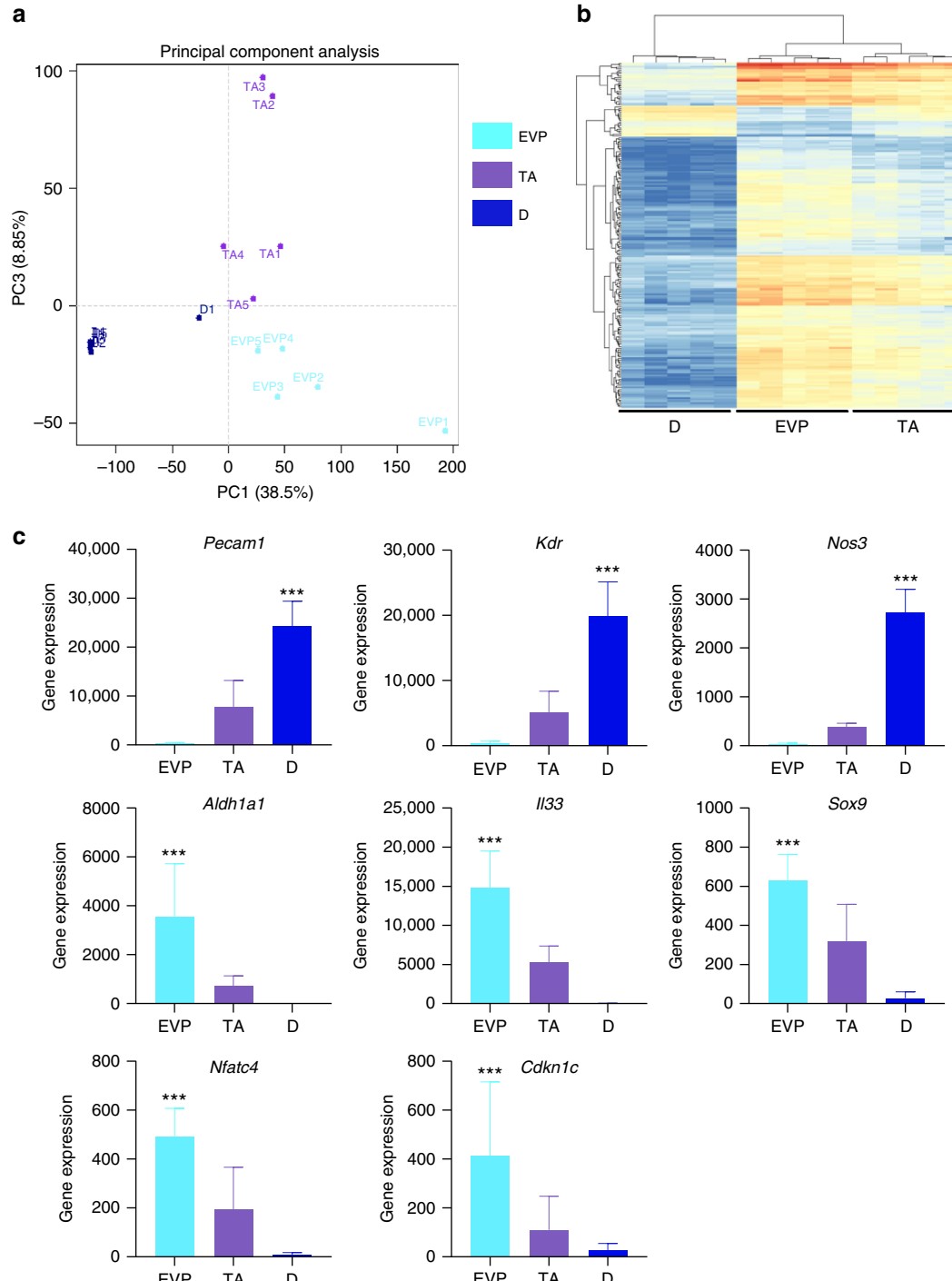

**Fig. 6** Endothelial hierarchy RNA sequencing and gene expression analysis. **a**, **b** Principal component analysis and hierarchical clustering demonstrating the distinct clustering of each population segregated from each other ($n = 5$). **c** Differentially expressed genes (*$p < 0.05$ after multiple testing correction) were identified between EVP and D populations. Differentiated endothelial markers *Pecam*, *Kdr*, and *Nos3* were all significantly upregulated in D compared to EVP (***$p < 0.001$ vs EVP). Results presented as mean ± SEM. EVP endovascular progenitor, D definitive differentiated

expression by plotting the fold change for overlapping genes in aorta vs tumors. Although there was a clear correlation in fold change between EVP and D in both aorta and tumors, some genes seemed to behave differently in the two situations. We therefore looked at genes that would specifically be overexpressed in EVP cells in the context of tumors and not aorta and identified 138 genes significantly overexpressed (at least 2 fold, $p < 0.05$, arrow in Supplementary Figure 5B). In the "tumor only" setting,

the most differentially upregulated genes in EVP were related to its paracrine function such as *Gdf6*, a member of the TGFβa agonist, *Wnt16*, *epiregulin*, and *neuroregulin*.

**Anti-VEGF-A therapy does not target EVP cells.** Having established the important role of EVP cells in tumor vasculature and its gene expression signature, we next wondered about strategies to target this population to reduce both tumor

**Table 1 Differentially expressed genes**

| Upregulated in EVP | Upregulated in D |
|---|---|
| **Proteases/ECM:** *Mmp* (13, 27, 19, 11, 9, 2, 10, 3, 23), *Dcr*, *Col3a1*, *Col1a2*, *Has1* | **Differentiated endothelial markers:** |
| **Angiocrine:** *Wnt* (9a, 5a, 10b, 16, 11, 2), *Fgf* (16, 23, 10, 11, 21, 2, 18, 7), *Hgf*, *Vegfd*, *Vegfa*, *Tgfβ3*, *Pdgfc* | *Pecam1*, *Kdr*, *Nos3*, *Vwf*, *Ets1*, *Ets2*, *Gata2*, *Fli1*, *Esam*, *Tie1*, *Cldn5*, *SoxF* (Sox7, 17, 18) |
| **Growth factors:** *Pdgfrα*, *Egfr*, *Tgfβr2*, *Tgfβr3* | **Notch signaling:** *Notch4*, *Notch 1*, *Dll4*, *Jag2*, *Hey1* |
| **Cytokine:** *Flt3*, *Il6ra*, *Il11ra1*, *Il1r11* | |
| **Cytokine targets:** *Socs1*, *Socs3*, *Ifi*, *Irf4*, *Irf6*, *Irf7*, *Pim1* | |
| **Stem cell markers:** *Aldh1a1*, *Sox9*, *Nfatc4*, *Il33*, *Cdkn1c* | |
| **Notch signaling:** *Notch3*, *Notch2*, *Rbpj*, *HeyL* | |

*EVP endovascular progenitors, D definitive differentiated*

vascularization, growth, and metastasis. Anti-VEGF-A therapy is the main FDA approved anti-angiogenic treatment for a range of solid tumors. However, it has failed in many indications to reduce tumor size, spread or vascularization and a variety of mechanisms have been proposed to explain resistance to this treatment[5]. We injected C57Bl/6 mice with B16-F0 melanomas and started therapy with anti-VEGF-A vs isotype control from D3 to D15 (Supplementary Figure 6A). Tumors were collected at D15 and analyzed for the relative frequency of endothelial populations: there was no significant reduction or change in the EVP or D population (Supplementary Figure 6B & C). This clearly shows that anti-VEGF-A therapy does not affect the pool of EVP cells available. There was no effect of anti-VEGF-A treatment on tumor vasculature in animals that received anti-VEGF-A therapy compared to control (Supplementary Figure 6D).

**Conditional ablation of Notch signaling alters EVP.** We have previously reported that active Notch signaling is essential in progenitor quiescence[23]. Moreover, RBPJ is a direct interactor with Sox18 suggesting that this pathway would be critical in EVPs. However, it has also been established that Notch signaling allows reduction of VEGFR2 receptors in sprouting angiogenesis forming stalk cells[24]. In our gene expression analysis, Notch signaling elements were overexpressed in D cells (*Notch4*, 47×; *Notch1*, 7×; *Dll4*, 73×; *Dll1*, 24×; *Jag2*, 17×) resulting in increased signaling via *Hey1* (13×), whereas in EVP cells other receptors were overexpressed (*Notch3*, 6×; *Notch2*, 16×; *Rbpj*, 4×) and signaling was mostly driven by *Heyl* (7×) suggesting that different ligand-receptors were involved in Notch signaling in different populations (Supplementary Figure 4E).

Given the indication that *Rbpj* as well as *Heyl* were overexpressed in EVP cells, we examined the consequences of endothelial-specific ablation of canonical Notch signaling. We used *Rbpj^{fl/fl}/Cdh5-Cre^{ER} RosaYFP* (*Rbpj^{eKO}*) mice and *Rbpj^{fl/fl} RosaYFP* (*Rbpj^{WT-Cre negative}*) controls (Fig. 7a, b). Animals were injected with tamoxifen for 10 days to fully ablate *Rbpj* in endothelial cells and they were inoculated with HCMel12 tumors intradermally. HCMel12 is a murine (C57Bl/6) derived melanoma cell line obtained carrying transgenic *Hgf* and *Cdk4r24c* mutation. Upon in vivo passaging this cell line is highly metastatic[25]. After 10 days, primary cutaneous tumors were excised and mice were kept alive to progress to metastasis (Fig. 7a). Primary tumors in *Rbpj^{eKO}* mice had dramatically smaller numbers of EVP cells whereas D cells were unchanged as compared to *Rbpj^{WT}* controls (Fig. 7b, c). In accordance, CD31 staining of primary tumor sections revealed only minimal difference between the two groups, whereas Lyve1 and Podoplanin staining of lymphatics was significantly increased in the absence of *Rbpj* (Fig. 7d). Of note in a B16-F0 model, primary tumors remained of smaller weight in *Rbpj^{eKO}* mice compared to

*Rbpj^{WT}* (Supplementary Figure 7; **p < 0.01; Mann–Whitney T-Test). At D28, 2 weeks after primary HCMel12 tumor excision, animals were sacrificed and examined for metastasis microscopically in the lung and the liver. We could not observe any significant metastasis in *RBPJ^{eKO}* animals whereas 50–90% of *Rbpj^{WT}* mice had identifiable tumors in lung and/or liver (Fig. 7e).

**Discussion**

Vascularization of tumors is a hallmark of cancer and has been shown as an important step in cancer progression and metastasis. Despite major progress in understanding key molecular pathways involved in angiogenesis, it remains unclear where tumor vasculature originates from and to what extent it has a supportive role in cancer progression. Here, we show that melanoma tumors from very early stages are infiltrated by a population of EVP. These cells initiate a vasculogenic process as a single cell and progress towards a TA stage and finally differentiate into arterial and venous capillaries within the central core tumor area. Of interest, a small fraction of these capillaries also gives rise to lymphatics although EVPs are clearly not derived from this vascular bed. At the molecular level, these progenitors are endothelial in origin as they express VE-Cadherin and reactivate the expression of the developmental transcription factor *Sox18*. RNA sequencing on sorted cells as well as functional characterization clearly differentiates EVPs from the later stages of progression towards differentiation. In particular, EVPs harbor significant paracrine activity that allows considering them as a reliable cellular target for therapy. Unlike anti-VEGF-A therapy, conditional ablation of *Rbpj* in the endothelium significantly reduced the EVP population in tumors and strongly inhibited metastasis in a model of melanoma.

Modes of new vessel formation have been described during development and include vasculogenesis and angiogenesis among others. Although our study does not address the contribution of angiogenesis, it clearly demonstrates the existence of a vasculogenic process within the tumor center that is devoid of any vessel. In this avascular area, single cells were tracked through their expression of either VE-Cadherin or Sox18 and shown to form entire blood vessels as already reported by us in skin wounds. This was further reinforced by multicolor lineage tracing showing that individual arterial or venous capillaries are formed by the juxtaposition of multiple clones and therefore unlikely to result from angiogenesis where a limited number of stalk cells would contribute to the clonal progression of vessel branching. In this setting, the vascularization process seemed to follow steps occurring during embryonic development through the expression of Sox18 but also the formation of lymphatics from cells that originate from different (Prox1 negative) vascular beds. Indeed adult lymphatic vessel formation has been shown to derive

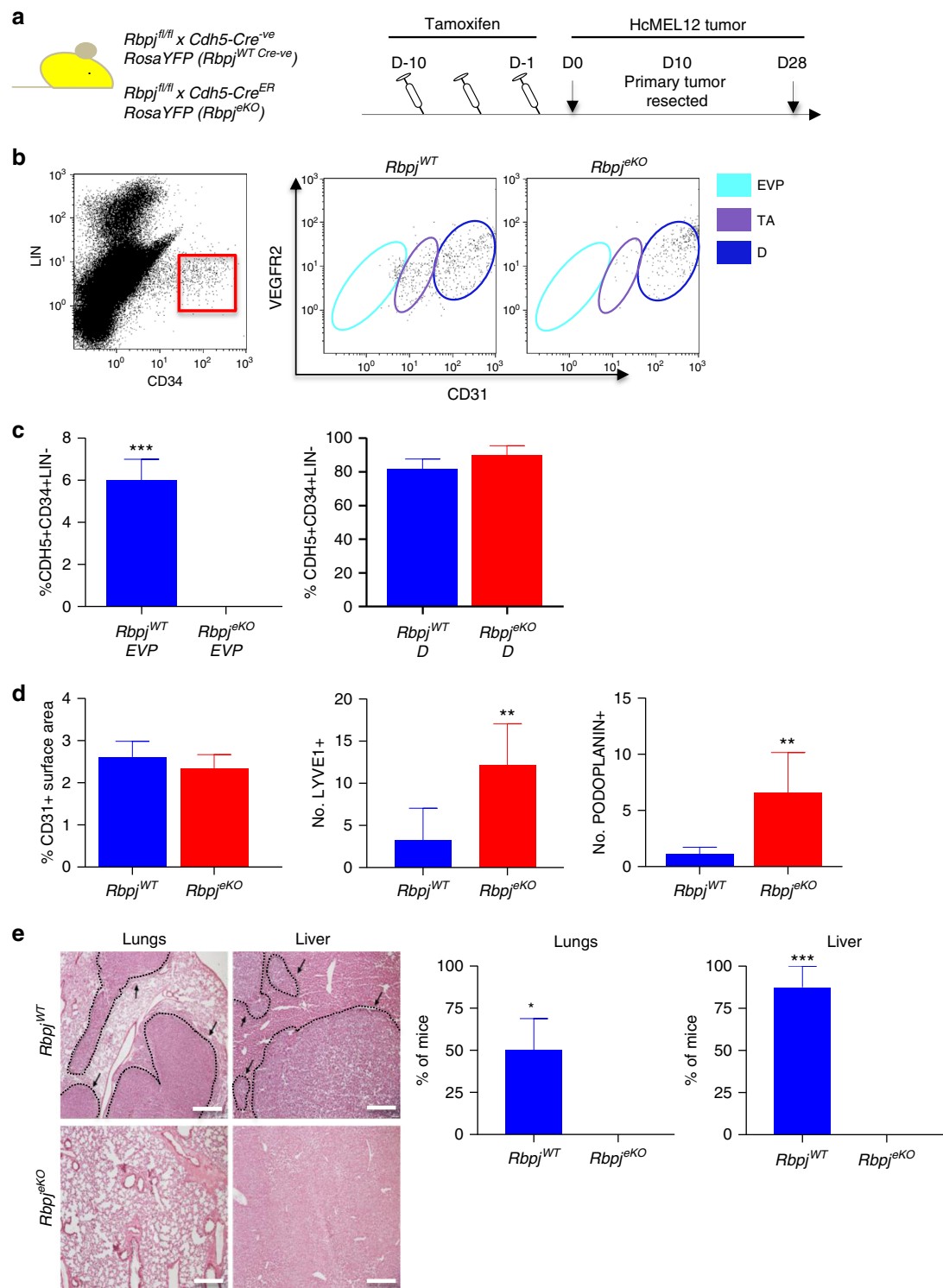

**Fig. 7** Vascular-specific knock-out of *Rbpj* results in EVP depletion and reduced metastasis. **a** Schematic diagram of the *Rbpj^fl/fl^/Cdh5-Cre^ER^ RosaYFP* (*Rbpj^eKO^*) and *Rbpj^fl/fl^ RosaYFP* (*Rbpj^WT-Cre Negative^*) lineage tracing model employed, with mice receiving tamoxifen (Tam) for 10 consecutive days prior to tumor inoculation with metastatic HcMEL12 melanoma cells injected at day 0 (D0). Tumors were harvested at D28 ($n = 5$). **b**, **c** Flow cytometry plots demonstrate that EVP cells are entirely absent from *Rbpj^eKO^* mice compared to *Rbpj^WT^* (***$p < 0.001$ vs *Rbpj^eKO^*; Mann–Whitney *T*-Test). No change was observed in the percentage of D cells present from either group. **d** No difference was observed in CD31+ vessel surface area between the groups. Significantly more Lyve1+ and Podoplanin+ vessels were observed in the *Rbpj^eKO^* mice compared to *Rbpj^WT^* (**$p < 0.01$ vs *Rbpj^WT^*; Mann–Whitney *T*-Test). **e** Histological analysis of lungs and liver showed vast metastatic nodules in the *Rbpj^WT^* compared to no visible nodules in the *Rbpj^eKO^* mice (*$p < 0.05$; ***$p < 0.001$ vs *Rbpj^eKO^*, Mann–Whitney *T*-Test). Results presented as mean ± SEM. Scale bar represents 150 μm. EVP endovascular progenitor, TA transit amplifying, D definitive differentiated

exclusively from existing lymphatics unlike the contribution of EVPs within the center of the tumor. Our findings cannot confirm or exclude if similar processes also occur at the periphery of the tumor. Indeed many vessels of blood or lymphatic origin surround the tumor. Given that the tumor periphery is not avascular, vasculogenesis is therefore unlikely in this area.

Progenitor activity of EVPs was exclusive and shown through their colony forming capacity as well as engraftment potential[8]. These properties could not be observed in TA and D populations. This strongly argues against the hypothesis that EVP, TA, and D are simply different stages of activation of otherwise similar endothelial cells. In the latter scenario where no hierarchy exists, one would expect equal engraftment and colony forming capacity. The observed functional differences suggest strongly that the cells differ intrinsically. This is strongly supported by the gene expression studies reported here clearly showing major transcriptomic changes from EVP to TA to D. Our findings highlight the transition from an endothelial cell with active expression of many mesenchymal genes important for mobility towards an endothelial cell that is adherent and expresses all of the endothelial differentiation markers including arterial and venous markers. We have previously shown a similar gene expression pattern within the aorta. The current findings reinforce the existence of a core gene expression signature that defines EVPs which include a capacity to remodel the extracellular matrix and a core set of "stem cell" genes and quiescence genes. Moreover, our findings add to lines of evidence supporting the concept of progenitors in the endothelium as defined by their function, self-renewal, plasticity, and lineage, rather than cell surface markers[9,10]. This hierarchy described in both mouse[11] and human[12] tissues allows a more precise definition of such progenitors. As such the current findings might differ from past studies where endothelial progenitors were defined based on a few surface markers without functional validation, an approach prone to contamination by hematopoietic or other cell lineages.

The impact of anti-angiogenic therapy on cancer progression is difficult to predict. There is even uncertainty on the goal of antiangiogenic therapy. Is it to normalize vessels to allow immune cell infiltration and drug delivery or is it to abrogate all vessel to prevent tumor growth and spread? We here also observe that EVPs have significant angiocrine activity producing many growth factors, morphogens, and cytokines that might affect tumor growth and spread. In support of this finding, inoculation of tumor cells with EVP compared to D cells resulted in significantly larger tumors. In such experiments, it is difficult to expect a large effect as the host is replete with EVPs and therefore the addition of external EVPs might not constitute a large advantage. The depletion of EVP cells through ablation of canonical Notch signaling as already observed in wounds and in vitro clearly showed the importance of these progenitors in tumor spread. Of note this did not significantly affect tumor CD31+ vessels. Previous studies of *Rbpj* depletion in the context of tumors have also reported an increase in immune cell infiltrate[26]. Interestingly, excision of tumors and follow-up showed clearly that this strategy was valid in neo-adjuvant settings. Of interest, in our hands the use of anti-VEGF-A therapy did not provide a similar protection and did not deplete EVP cells. Indeed anti-VEGF-A is unlikely to target EVP cells given their low level of expression of VEGF receptors. Given the demonstrated continuous input of progenitors we propose here that EVPs are probably a major therapeutic target for cancer control.

In conclusion, tumor vascularization recapitulates mechanism observed during embryonic development via the activation and expression of sox18 in specific progenitors of endothelial origin that form arterial and venous capillaries but also lymphatics. The hierarchy between progenitor, TA, and differentiated cell can be tracked by fate tracing as well as gene expression studies. Our findings strongly suggest that targeting EVPs is a valid strategy in controlling the progression and spread of cancer.

## Methods

**Animals.** All mice were treated in accordance with University of Queensland ethics approvals and guidelines for care of experimental animals. Both males and females (8–14 weeks of age; genders housed separately) were used for this study. C57BL/6 mice (WT) were obtained from the Animal Resources Centre (Perth, Western Australia). For lineage tracing experiments, *Sox18-Cre^ERt2* and *Cdh5-Cre^ERt2* were crossed with ROSA_lox_YFP_lox. The resultant double transgenic offspring were named *Sox18-Cre^ER RosaYFP* (8–12 weeks old) and *Cdh5-Cre^ER RosaYFP*. For endothelial-specific knockout of the gene *Rbpj*, *Rbpj^fl-fl* mice were crossed with *Cdh5-Cre^ER RosaYFP* to create the resultant triple-transgenic *Rbpj^fl/fl/Cdh5-Cre^ER RosaYFP*. For polyclonal lineage tracing, *Cdh5-Cre^ERt2* were crossed with the *Rainbow* (CAG-Brainbow 1.0) mice to create the resultant *Cdh5-Cre^ER Rainbow*. *Prox1-Cre^ERt2* were crossed with *tDTomato* reporter mice to create the resultant *Prox1-Cre^ER tDTomato*. Ubiquitous *CAG-GFP* mice were used for all stem cell transplant experiments. All lineage tracing experiments were conducted using Tamoxifen (Sigma-Aldrich, MI, USA) made up in 90% corn oil (Sigma-Aldrich) and 10% ethanol, with each mouse receiving a 2 mg dose per intraperitoneal injection (100 μL of 20 mg/ml solution). Mice being treated with either anti-mouse VEGF-A (BioLegend, #512808, CA, USA) or Isotype control (BioLegend #400533)) were given a dose of 100 μg every 4 days via intraperitoneal injections.

**Tumor cell culture.** B16-F0 cells were cultured using RPMI 1640 supplemented with 10% fetal bovine serum (FBS). Cells were passaged every 4 days, with $5 \times 10^5$ cells in saline (300 μL) were injected subcutaneously on each flank of mice for in vivo tumor studies. Metastatic melanoma cell line HcMel12 cells were obtained as a single cell suspension following 3 subsequent in vivo passaging (in vivo tumors were passaged after 14 days of growth). $2 \times 10^5$ cells in saline (300 μL) were injected subcutaneously for in vivo tumor studies.

**Tissue processing of murine B16-F0 and HcMel12 tumors.** Tissues were collected for ex vivo analyses at defined end points tumors (D5, D10, and D15). Tumors were first digested for 20 min at 37 °C in 1 mg/ml collagenase I (Gibco, Life Technologies, NY, USA), 1 mg/ml dispase (Gibco, Life Technologies, NY, USA), 150 μg/ml DNase-I (Sigma-Aldrich, St. Louis, MO, USA) before passing the suspension through a 70 μM cell strainer. Lineage+ cells were then depleted from tumor cell suspensions via MACS® cell separation according to the manufacturer's instructions (Miltenyi Biotech, Cologne, Germany). Cell number and viability for each sample was assessed using 0.4% Trypan blue solution and a hemocytometer. Single cell suspensions were then used for flow sorting or analysis by flow cytometry. For transplantation studies, tumors were grown in CAG-GFP+ mice and GFP+ EVP, TA, and D cells were isolated from harvested tumors.

**Flow cytometry and FACS.** Dissociated single cells in PBS/BSA/EDTA were then incubated with various antibody combinations for multi-parameter flow acquisition and analysis. A Gallios™ flow cytometer was used for sample acquisition, while unbiased data analyses were performed with Kaluza® analysis software (Beckman-Coulter, Miami, FL, USA). FACS was performed by using a FACSaria cell sorter (Becton Dickinson, Franklin Lakes, NJ, USA). Extreme care was taken during cell sorting to ensure only "singlets" were being gated and any potential "doublets" were being gated out. Cell populations were collected in 5 ml polypropylene tubes containing 100% fetal calf serum (FCS). The following combinations of antibodies were used to assess the endothelial hierarchy populations: Rat anti-mouse VEGFR2 PE (1:50), Sca-1 (1:50), c-KIT (1:50), CD31 PE-Cy7 (1:100), and CD34 Alexa647 (1:50) (Becton Dickinson, NJ, USA), Rat anti-mouse Lineage cocktail BV450 (1:50) (BioLegend), Rat anti-mouse CD144 FITC (1:50) (eBioscience). FMO was used to delineate negative gating for each antibody.

**RNA extraction.** RNA was extracted from FACS sorted EVP, TA, and D cells using a QIAGEN mini kit (Qiagen, Valencia, CA) according to the manufacturer's instructions. RNA quality and concentration was assessed using A260 nm/A280 nm spectroscopy on the Nanodrop ND-1000 (Thermo-scientific, Langenselbold, Germany). 5–100 ng of RNA was used for cDNA synthesis using the Superscript III Reverse Transcription Kit (Invitrogen, Mount Waverley, Australia).

**Library preparation, RNASeq, and data analysis.** RNA-Seq libraries were prepared from purified total RNA using a modified Smart-Seq2 protocol[27]. 2 ng of purified total RNA (0.4 ng/μL) was combined with 1 μL of 10 μM oligo-dT primer (/5Biosg/AAGCAGTGGTATCAACGCAGAGTACT30VN; Integrated DNA Technologies) and 1 μL of dNTP mix (10 mM each; Invitrogen, y02256), then the protocol was continued as described (ref. [2]). Briefly, the RNA was reverse transcribed with the Smart-Seq2 TSO (/5Biosg/AAGCAGTGGTATCAACGCAGAG TACATrGrGrG, Integrated DNA Technologies), followed by 12 cycles of PCR amplification to obtain enough cDNA to prepare a library. Volumes of reagents

were scaled accordingly to maintain final concentration ratios as in the original protocol, except for the PCR preamplification where the Smart-Seq2 ISPCR primer (/5Biosg/AAGCAGTGGTATCAACGCAGAGT; Integrated DNA Technologies) was added to a final concentration of 0.25 μM. 0.5 ng of cDNA was prepped into a library using the Nextera XT DNA sample preparation kit (Illumina, FC-131-1096), with 12 cycles of PCR used to amplify the final library. The final Nextera XT libraries were quantified on the Agilent Bioanalyzer with the High Sensitivity DNA kit (Agilent Technologies, 4067–4626). Libraries were pooled in equimolar ratios, and the pool was quantified by qPCR using the KAPA Library Quantification Kit—Illumina/Universal (KAPA Biosystems, KK4824) in combination with the Life Technologies Viia 7 real time PCR instrument. Sequencing was performed using the Illumina NextSeq500 (NextSeq control software v2.0.2/Real Time Analysis v2.4.11). The library pool was diluted and denatured according to the standard NextSeq protocol (Document # 15048776 v02), and sequenced to generate paired-end 76 bp reads using a 150 cycle NextSeq500/550 High Output reagent Kit v2 (Catalog # FC-404-2002). After sequencing, fastq files were generated using bcl2fastq2 (v2.17). Library preparation and sequencing was performed at the Institute for Molecular Bioscience Sequencing Facility (University of Queensland).

**Data analysis**. The RNA-seq reads were mapped to the Mus musculus mm10 genome with STAR version 2.5.3a[28] using Ensembl annotation (GRCm38, release 91). Reads were quantified with HTSeqcount version 0.6.1[29] using the attribute gene_id from the Ensembl GTF file GRCm38 release 91 as feature ID. Read count normalization and differential gene expression analysis was performed using DESeq2 version 1.10.1[30].

**Immunofluorescence and imaging**. Dissected tumors were fixed for 2 h in 4% PFA. The fixative was removed with 3× washes of 1× PBS (Amresco, Solon, OH, USA). Tissues were subsequently infused with sucrose before cryo-embedding. For staining of specific antigens, cryo-sections were permeabilized in 0.5% Triton-X-100 (Chem Supply, Gillman, South Australia) before blocking with 20% normal goat serum. For this study, primary antibodies used included rat anti-mouse CD31 (1:50), rat anti-mouse VEGFR2 (1:50), rat anti-mouse CD34 (1:50) (all from Becton Dickinson); rabbit anti-mouse LYVE-1 (1:100), rabbit anti-mouse Dll4 (1:250), rabbit anti-mouse endomucin (1:100) (Abcam, MA, USA) and hamster anti-mouse Podoplanin (1:100) (AngioBio, CA, USA). Excess and unbound antibody was then removed with 3 × 5 min washes in a solution containing 1× PBS/0.1% Tween-20 (Amresco, Solon, OH, USA). Secondary antibodies conjugated with Alexa-Fluor 568 or 647 (Invitrogen, Carlsbad, CA, USA) were used for fluorescence detection. Briefly, sections were incubated with secondary antibodies for 40 min at room temperature. Excess antibody was removed by 3× washes in PBS/0.1% Tween-20. Nuclear staining was revealed in specimens mounted with ProLong® Gold mounting media containing DAPI (Invitrogen, Carlsbad, CA, USA). Confocal images were acquired with a Zeiss LSM 710 microscope equipped with Argon 561-10 nm DPSS and 633 nm HeNe lasers, and a 405-30 nm diode. Images were obtained at 10× and 20×. Immunofluorescence vessel quantification was conducted using ImageJ (NIH). Vascular beds were located and imaged as described above. The region of interest was kept consistent, and vascular beds were overlaid with the YFP channel and quantified as a percentage overlap standardized to YFP coverage.

**Statistical analysis**. All statistical analyses were performed using GraphPad Prism v5c software. Data were analyzed using the following tests: Mann–Whitney (for non-parametric data), $T$-tests 1-way or 2-way ANOVA with Bonferroni correction for parametric data. A $p$-value < 0.05 was considered significant.

**Reporting summary**. Further information on experimental design is available in the Nature Research Reporting Summary linked to this article.

## Data availability

All sequencing data that support the findings of this study have been deposited in the NCBI Gene Expression Omnibus (GEO) and are accessible through the GEO Series accession number GSE114528.

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

## Acknowledgements

The authors would like to thank Dr. Graeme Walker and Dr. Tobias Bald for access to murine models of melanoma. The study was funded by the NHMRC Project Grant (APP1085142). K.K. and M.F. salary was supported by NHMRC Career Development Fellowship (APP1125290 and APP1111169). J.P. salary was supported by the ARC DECRA Research Fellowship (DE180100984). K.K. and M.F. secured funding and management for the study.

## Author contributions

P.D. and J.P. designed and conducted experiments and assembled figures and manuscript. J.D., H.Y.W., S.-L.S. and V.M. provided assistance during experimentation. M.F. provided assistance in study design and manuscript preparation. J.P., M.F. and K.K. designed the study and assisted in assembling the figures and manuscript and finalizing the manuscript.

## Additional information

**Competing interests:** J.P. and K.K. are co-inventors on a patent relating to the isolation of endothelial progenitors from the placenta. The other authors declare no competing interests.

