## [Peer Review File · Nature Communications]

Reviewers' comments:

Reviewer #1 (Remarks to the Author):

This study describes the development of tumor vascularization. This vascularization is initiated by the presence of endovascular progenitors (EVP). These EVP differentiated into transit amplifying (TA) and differentiated (D) populations. These EVPs could give rise to venous, arterial and lymphatic vascular beds. The EVPs reactivated the expression of SOX18 transcription factor, recapitulating the mechanisms observed in embryonic development. Although the findings are interesting and possibly suited for the readership of Nature Communications, the manuscript seems to require further maturation (see specific points below).

Major comments

1. The authors mention in the title, and in the manuscript, that endovascular progenitors infiltrate melanoma tumors. The results show the presence of these EVP cells at different time points, however it is not shown if these cells infiltrate or if they were already present (see for instance <https://www.ncbi.nlm.nih.gov/pubmed/28650345>), or if they have been differentiated from another precursor cell type. Additional experiment to rule out the pre-existence, or to prove the infiltration of the EVPs would be highly appreciated.
2. The authors refer several times in the manuscript (lines 63,188,196,200,212,233,358, 433, 451) that they have reported similar findings previously. This raises the question if the presented findings in this manuscript show sufficient novelty.
3. The analyses were validated in a spontaneous melanoma model (line 198, Suppl. Fig. 1B). Could this model not be used to confirm more of the presented data? Otherwise, could another tumor type be included to show that the observed results are generally applicable?
4. L216- , Fig 2.A+B: Are YFP+CD34+ cells only formed in the context of a tumor? Or are they also present in healthy skin tissue after tamoxifen administration in these mice? Please show some experimental data with these control tissues.
5. L37: Rbpj depletion by gene knockout has revealed the mechanism, but can the authors show some data that this understanding has the potency to result in therapy? Line 476; have the authors arguments to support their claim that "therapies that target EVPs are more likely to be effective in controlling progression and spread of cancer"?

Ref 28650345: Preexisting endothelial cells mediate cardiac neovascularization after injury. J Clin Invest. 2017

Minor comments

1. At line 430, the authors state “although our study does not address the contribution of angiogenesis”. However, the manuscript starts with “tumor angiogenesis...”. To me this is confusing. Maybe elaborate more on the different types of tumor vascularization?
2. Lines 42-45: What is the main message of this paragraph? Is tumor angiogenesis pro- or anti-tumorigenic?
3. Lines 67-69: Here is stated that EVPs form venous and lymphatic vessels. But also arterial vessels are formed (see line 252).
4. Line 73: The authors mention that ablation of RBPJ has a result on metastasis. Is there also a result on tumor growth? If not, could the authors provide some arguments why this is not the case?
5. Which gender of mice was used? How were the animals housed?
6. L87: How was tamoxifen administered (volume, dose/mice)? Which tamoxifen was used?
7. L88: Which anti-mouse VEGF-A has been used (catalog. No, company)?
8. L92: Were the B16-FO cells authenticated (same for all other cell lines)? How were the tumors initiated (medium, volume)?
9. L94: Regarding to in vivo passage: When were the cells passaged (was this based on tumor volume or at a fixed time point)
10. L118-120: Were isotype controls included in the FACS experiments?
11. L131+134: What are refs 1+2?
12. In the methods section is not explained how the vascular beds are quantified (e.g. for Figs. 3+4)
13. All Figures: The numbering of the figures is cropped of the pages.
14. L207: “Only EVP cells were able to persis and engraft and from endothelium ..”. How do the FACS data form Figure 1 show that endothelium is formed?
15. All FACS figures: Quantification is missing.
16. Fig 1. C, D: Have the GFP+ EVP’s expanded in number? Or do these data only show that this cell population stays alive (in contrast to TA and D cells)? What is expressed on Y-axis of 1D? Percentage or number of cells? What are the CON cells?
17. L209-211: The differences between the groups seem not statistically different; therefore, the conclusion that tumor cells delivered with EVP results in large tumors whereas tumor cells delivered with D cells in small tumors cannot be made. Information on the number of mice and at which time the tumors were evaluated is missing. Error bar for CON group is missing.

18. L224-230: Text states that at D15, D cells were the only remaining population. But Fig 2A, %YFP+ hierarchy shows approximately 5% EVP cells (even an increase to D10) and ~20% TA cells at D15. Please explain this discrepancy between text and figure.
19. L237- : Text says: "At this time point [D5] we could not observe any TA or D cells. (Fig 2C, D)", but in the figure there are dots visible in the gates TA cells. Line 239 says that D populations could be observed D10-D20 post-tumor inoculation. Perhaps this should be D10-D15?
20. L244-250: What is the origin of the 2nd wave of EVPs?
21. Figure 3: At D10, are the CDH5 cells still single cells, or part of the vessels? Larger magnification figures, and single channel figure could be helpful.
22. Figure 4A: Is it correct that upper and lower panels both display CDH5?
23. Figure 5B: In the left panel, many purple dots (nuclei?) are visible, in the right panel these are not visible anymore. Have they disappeared? Is magnification of the right panel the same as the others?
24. Line 305: "The three populations could be easily distinguished. (figure 6A)", in my opinion the populations are not easily to distinguish. Most samples fall into the area within -10 – 90% PC1 and -30 – 20% OC3.
25. More than two pages (line 316 – 394) are used to describe supplementary figures only. Either include main figures or shorten the text.
26. Line 372-383: Why is the anti-VEGF-A therapy only presented as supplementary figures? From a clinical perspective, this is interesting data.
27. Figure 7E: The error bars suggest that up to 125% of the mice have liver metastasis.

Reviewer #2 (Remarks to the Author):

Tumor angiogenesis has been a major research direction in the last few decades. The majority of studies focused on sprouting angiogenesis from nearby existing blood vessels that contribute to tumor angiogenesis. However, over the years, there has been a substantial debate regarding the contribution of endothelial progenitor cells to tumor angiogenesis.

In the current study, Donovan et al., provide another piece of important information which clearly demonstrate how bone marrow derived endovascular progenitors (EVP) contribute to blood vessel formation in tumors. Using several unique genetically modified mouse systems along with traceable lineage differentiation of progenitor cells, the authors elegantly demonstrate that vasculogenesis is a major angiogenesis path in cancer.

This is a thorough study with extensive details as for how vasculogenesis takes place during tumor growth. The majority of the results support the conclusions of the study. The experimental plan was rationally designed. I therefore believe that this manuscript is suitable for publication in Nature Communication. Yet, I outlined below several drawbacks that need to be addressed prior to publication.

Critique

1. Materials and Methods: LYVE1 is expressed on macrophages as well as other cell types and not only lymphatic endothelial cells. It is better to immunostain with podoplanin. I noticed that the authors also used PROX1, but it seems that it was not mentioned in the Materials and Methods.
2. Page 9: Can the authors demonstrate that endovascular progenitor cells (EVP) in which Sox18 maintained at higher levels do not tend to differentiate. The experiment performed is correlative, but do not provide a full direct proof that Sox18 regulates differentiation of EVPs.
3. It is not clear whether the lymphatic vessels YFP+ are true lymphatic vessels or they can be other cell types, as LYVE1 is expressed by different cells. Staining with Podoplanin (as indicated above) is essential to evaluate the contribution of EVPs to lymphatic endothelial cells.
4. How do the authors know that the use of Lin-CD34+/VEGFR2+ are not from the hematopoietic lineage. Can the authors demonstrate that such cells do not express Sca1 and other hematopoietic progenitor markers (CD152, etc)?
5. Anti-VEGF-A is indeed the main antiangiogenic therapy approved drug. Yet, other small molecule drugs have been approved for several malignancies, e.g., sunitinib, sorafinib etc. These drugs are tyrosine kinase inhibitors that block several pathways of angiogenesis. It would be of interest to evaluate whether these small molecule drugs affect EVPs as opposed to anti-VEGF-A.
6. Several studies argued that the contribution of endothelial progenitor cells to tumor growth and metastasis is minimal if at all. This has not been discussed in the context of this study, and it should as this area of research is under heavy debate.
7. Figure 2B, can the authors immunostain CD31 blood vessels. In other words, how many of the vessels in the tumor are YFP+ and how many of them are angiogenesis per se (not vasculogenesis)?
8. Figure 3D, there is overexposure of pixels in the merged figure of Lyve1/Prox1.

Reviewer #1

- The authors mention in the title, and in the manuscript, that endovascular progenitors infiltrate melanoma tumors. The results show the presence of these EVP cells at different time point, however it is not shown if these cell infiltrate or if they were already present (see for instance <https://www.ncbi.nlm.nih.gov/pubmed/28650345>), or if they have been differentiated from another precursor cell type. Additional experiment to rule out the pre-existence, or to prove the infiltration of the EVPs would be highly appreciated.

We thank the reviewer for this comment. Indeed, this is an important and difficult question as it is nearly impossible to rule out contribution from a distant source to the primary tumor. In light of previous work (Patel et al. *Circulation* 2017)¹ we know that EVPs are not derived from the bone marrow or at least they cannot be transplanted. This is in support of a local source of EVPs. Moreover, given the rate at which these tumor models grow, they often have a center that is devoid of any vasculature. Therefore the presence of EVP cells in these areas is likely to derive from the surrounding vasculature infiltrating the tumor. We have now discussed this in more detail.

However, to fully convince ourselves of the strict endothelial origin of EVPs supporting their infiltrative nature we have conducted the following label dilution experiments as performed before (He et al. *JCI* 2017)². Indeed we assumed that if a source other than the endothelium was giving rise to EVP and subsequently to the vasculature, less and less endothelial cells would be labelled over time during lineage tracing.

We show in the new data (Supplementary Figure 2B-D; Manuscript Text Page 10) that upon lineage tracing using a *Cdh5* promoter, the endothelial compartment in the back skin after tamoxifen injection are >99% YFP+. We then show similar levels (>99%) of labelling in mice induced at the same time and inoculated with B16F0 tumors two weeks later and finally collected 24 days after their final tamoxifen injection. This clearly demonstrates that the entire tumor vasculature is derived from pre-existing endothelium and that no other cell type (not expressing VE-Cadherin) has contributed to tumor endothelium. Overall these new findings further confirm the origin of EVPs and support their infiltration of tumors.

- The authors refer several times in the manuscript (lines 63,188,196,200,212,233,358, 433, 451) that they have reported similar findings previously. This raises the question if the presented findings in this manuscript show sufficient novelty.

We thank the reviewer for this comment. We have addressed this in various locations in the manuscript to reduce repetition of statements. However, we would like to point out, that our previous publication in *Circulation* (Patel et al. *Circulation* 2017) provided the

field clarity in how “endothelial progenitors” are assessed in the vascular compartment. Therefore, referencing this assessment is critical at certain points in the text to ensure this definition is being used appropriately. This does not reduce the novelty of our current findings, which demonstrate the plasticity and functional importance of EVP in a tumor setting.

- The analyses were validated in a spontaneous melanoma model (line 198, Suppl. Fig. 1B). Could this model not be used to confirm more of the presented data? Otherwise, could another tumor type be included to show that the observed results are generally applicable?

We thank the reviewer for this comment. To address this we conducted additional lineage tracing experiments using 2 cancer cell lines, Lewis Lung and the EO771 Breast Cancer lines (Supplementary Figure 1D; Manuscript Text Page 9). Solid tumors grown subcutaneously from these cell lines for 14 days also resulted in the observation of the same endothelial hierarchy as observed with the tested melanoma cell lines as well as the spontaneous assessed. We are very confident in the reproducibility of our data amongst various organ beds and cancer types.

- L216- , Fig 2.A+B: Are YFP+CD34+ cells only formed in the context of a tumor? Or are they also present in healthy skin tissue after tamoxifen administration in these mice? Please show some experimental data with these control tissues.

We provide now evidence that the YFP+CD34+ populations, EVP, TA and D are present also in normal skin. This has been addressed in comment #1 (Supplementary Figure 2C). Of importance, this is consistent with our findings in the normal aorta and normal lung as published previously.

- L37: Rbpj depletions by gene knockout has revealed the mechanism, but can the authors show some data that this understanding has the potency to result in therapy? Line 476; have the authors arguments to support their claim that “therapies that target EVPs are more likely to be effective in controlling progression and spread of cancer”?

We understand the reviewer’s question about translational potential of our findings. Although this is an important question, we believe it is beyond the aims of the present study. Even if we have not yet produced any data in support of this, there are a variety of Notch pathway inhibitors tested to affect tumors directly or indirectly. We have previously shown in an *in vitro* setting that DAPT, a small molecule inhibitor of canonical NOTCH signaling inhibits endothelial progenitor self-renewal. Similar products, such as CB-103 from Cellestia, are in development (phase II trials,

<https://www.cellestia.com/product/pipeline>) and allow targeting Notch related transcription in a similar way as the RBPJ inhibition.

Overall, many of the anti-angiogenic therapies described in the literature such as anti-VCAM therapy, anti-CD248 therapy more specifically target the EVP than other populations. However, we have now toned down our discussion and removed the claim that therapies that target EVPs are more likely to be effective as we have not formally tested this.

Minor comments

- At line 430, the authors state “although our study does not address the contribution of angiogenesis”. However, the manuscript starts with “tumor angiogenesis...”. To me this is confusing. Maybe elaborate more on the different types of tumor vascularization?

We agree with reviewer that our study focuses more clearly on a vasculogenic process occurring in the center of tumors. We have now changed this sentence to “tumor “vascularization”. We have also discussed this in more depth and separate our observations from the classical angiogenesis that is probably occurring in the tumor surroundings”.

- Lines 42-45: What is the main message of this paragraph? Is tumor angiogenesis pro- or anti-tumorigenic?

We apologize if the intent of this paragraph was unclear. Indeed tumor vessels have both detrimental roles (tumor cell spread, provision of oxygen and nutrient, poor oxygenation maintaining an invasive state) as well as beneficial roles (immune cell and drug delivery). This has been amended in the text on page 3.

- Lines 67-69: Here is stated that EVPs form venous and lymphatic vessels. But also arterial vessels are formed (see line 252).

This has been amended in the text on page 3.

- Line 73: The authors mention that ablation of RBPJ has a result on metastasis. Is there also a result on tumor growth? If not, could the authors provide some arguments why this is not the case?

Conditional ablation of RBPJ also resulted in a reduction in primary tumor size when repeated with B16-F0 tumors. Indeed these experiments could not be performed with the HCMel12 as primary tumors are resected after 10 days. This is now corrected in

Figure 7A. The new findings about primary tumor size is now added in Supplementary Figure 7 material. This has been amended in the text on page 15 and 23.

- Which gender of mice was used? How were the animals housed?

We used both males and females throughout the studies. They were housed according to gender, with 2 mice per cage. This has been amended in the text on page 5.

- L87: How was tamoxifen administered (volume, dose/mice)? Which tamoxifen was used?

Tamoxifen was injected intraperitoneally at a dose of 2mg/day (100 μ L solution; tamoxifen/corn oil/ethanol). This has been clarified in the text on page 5.

- L88: Which anti-mouse VEGF-A has been used (catalog. No, company)?

This has been clarified in the text on page 5.

- L92: Were the B16-F0 cells authenticated (same for all other cell lines)?

B16-F0 cell line is mouse derived tumor line. We have made very similar observations with a variety of tumor lines and do not assume any specific variable related to the tumor itself would modify our observations on tumor environment. As such we do not believe further authentication is warranted.

- How were the tumors initiated (medium, volume)?

This has been clarified in the text on page 5.

- L94: Regarding *in vivo* passage: When were the cells passaged (was this based on tumor volume or at a fixed time point).

This has been amended in the text on page 5.

- L118-120: Were isotype controls included in the FACS experiments?

This has been amended in the text on page 6.

- L131+134: What are refs 1+2?

This has been amended in the text on page 6.

-
- In the methods section its not explained how the vascular beds are quantified (e.g. for Figs. 3+4).

This has been amended in text on page 7.

- L207: “Only EVP cells were able to persist and engraft and form endothelium”. How do the FACS data form Figure 1 show that endothelium is formed?

We now provide additional data demonstrating the expression of CD34 and the absence of lineage markers in this population. This has been amended in the text on page 9.

- All FACS figures: Quantification is missing.

We respectfully disagree with reviewer. For every FACS plot that is subjected to a comparison we have provided bar graphs with precise quantification and standard deviation of repeat experiments. (Figure 1D, Figure 2A&C (kinetics curve of quantities), and Figure 7C). The only FACS plot not presenting quantification is on Figure 4E, which demonstrates that no EVP cells derive from Prox1 expressing lymphatics.

- Fig 1. C, D: Have the GFP+ EVP's expanded in number? Or do these data only show that this cell population stays alive (in contrast to TA and D cells)? What is expressed on Y-axis of 1D? Percentage or number of cells? What are the CON cells?

This has been amended in the text on page 9 and 20. The process of transplantation results in significant death and then re-expansion of engrafted cells as shown by us previously (Shafiee et al, FASEB J 2018)³. 1D reflects the number of cells recovered. We further provide sections of these tumors with GFP+ engrafted EVPs in Figure 1E.

- L209-211: The differences between the groups seem not statistically different; therefore, the conclusion that tumor cells delivered with EVP results in large tumors whereas tumor cells delivered with D cells in small tumors cannot be made. Information on the number of mice and at which time the tumors were evaluated is missing. Error bar for CON group is missing.

We have now provided additional experiments to demonstrate significant differences in tumor size between tumors engrafted with D cells versus those engrafted with EVP cells. EVPs enhanced tumor growth significantly. To ensure that the comparison is valid across the 4 experiments performed, in each series tumor size has been reported to the

control group (therefore serving as a reference tumor growth, 1). Tumors enhanced with EVPs grew in average 50% more and those with D cells grew about 30% less. This has been amended in the text on page 9 and 20 and Figure 1F.

- L224-230: Text states that at D15, D cells were the only remaining population. But Fig 2A, %YFP+ hierarchy shows approximately 5% EVP cells (even an increase to D10) and ~20% TA cells at D15. Please explain this discrepancy between text and figure.

This has been amended in the text on page 10.

- L237- : Text says: “At this time point [D5] we could not observe any TA or D cells. (Fig 2C, D)”, but in the figure there are dots visible in the gates TA cells. Line 239 says that D populations could be observed D10-D20 post-tumor inoculation. Perhaps this should be D10-D15?

This has been amended in the text on page 11. At this time point TA and D cells were nearly absent.

- L244-250: What is the origin of the 2nd wave of EVPs?

We believe all EVP waves originate from the surrounding vasculature. The EVP waves are probably continuous and emanate from the same endothelial source infiltrating the tumor.

- Figure 3: At D10, are the CDH5 cells still single cells, or part of the vessels? Larger magnification figures, and single channel figure could be helpful.

We thank the reviewer for this comment. We provide numerous example showing that the EVPs at D10 have already formed vascular structures when emanating from CDH5 expressing cells. An example is Figure 5 with higher magnification.

- Figure 4A: Is it correct that upper and lower panels both display CDH5?

This has been corrected on the figure to say CDH5 (upper) and Sox18 (lower)

- Figure 5B: In the left panel, many purple dots (nuclei?) are visible, in the right panel these are not visible anymore. Have they disappeared? Is magnification of the right panel the same as the others?

The purple reflects DAPI nuclear staining. The image on the right of the panel is at a higher magnification, this has been clarified in the text on page 21.

-
- Line 305: “The three populations could be easily distinguished. (figure 6A)”, in my opinion the populations are not easily to distinguish. Most samples fall into the area within -10 – 90% PC1 and -30 – 20% OC3.

This has been amended in the text on page 13.

- More than two pages (line 316 – 394) are used to describe supplementary figures only. Either include main figures or shorten the text.

We have provided important descriptive information about the gene expression differences occurring between EVP, TA and D. Given the three party comparison it is difficult to provide simple summary data. However, if the editor/reviewer think this is excessive we can accommodate part of this description in supplemental data.

- Line 372-383: Why is the anti-VEGF-A therapy only presented as supplementary figures? From a clinical perspective, this is interesting data.

We agree with reviewer. However, we are subjected to editorial limitations. Also the use of anti-VEGF is not the main point of this study but a comparison point.

- Figure 7E: The error bars suggest that up to 125% of the mice have liver metastasis.

This has been amended in Figure 7.

Reviewer #2

- LYVE1 is expressed on macrophages as well as other cell types and not only lymphatic endothelial cells. It is better to immunostain with podoplanin. I noticed that the authors also used PROX1, but it seems that it was not mentioned in the Materials and Methods.

We thank the reviewer for this comment. To address this we have conducted additional staining with Podoplanin and conducted additional quantification with this marker. This has been amended in Figures 4, 7 and Supplementary Figure 6.

- Page 9: Can the authors demonstrate that endovascular progenitor cells (EVP) in which Sox18 maintained at higher levels do not tend to differentiate. The experiment performed is correlative, but do not provide a full direct proof that Sox18 regulates differentiation of EVPs.
-

Sox18 is a transcription factor that is usually absent from adult endothelium and becomes expressed during wound or cancer vascularization. In our previous work, functional loss of Sox18/SoxF resulted in the persistence of EVPs and a reduction of TA and D cells in wounds. In this work, RNA levels of Sox18 are higher in D than in TA and EVP. Lineage tracing also shows that Sox18 expressing cells irretrievably progress towards TA and D (Figure 2C). Overall, we believe that Sox18 expression is initiated in EVPs and increased to promote their differentiation towards TA and D cells.

- How do the authors know that the use of Lin-CD34⁺/VEGFR2⁺ are not from the hematopoietic lineage. Can the authors demonstrate that such cells do not express Sca1 and other hematopoietic progenitor markers (CD152, etc)?

We thank reviewer for this question. It is important to keep in mind that these experiments are done with subcutaneous tumors and there is no hematopoietic tissue and specifically no bone marrow involved. Furthermore, we have now provided additional evidence (Supplementary Figure 1B) that among EVPs, there is no c-KIT⁺ cells, however all endothelial populations are expectedly SCA1⁺. Overall, there is no indication that primitive hematopoietic progenitors that would be LIN⁻ are circulating in the subcutis, and even if this was the case, EVPs do not express some of the common markers of these progenitors such as c-KIT.

- Anti-VEGF-A is indeed the main antiangiogenic therapy approved drug. Yet, other small molecule drugs have been approved for several malignancies, e.g., sunitinib, sorafenib etc. These drugs are tyrosine kinase inhibitors that block several pathways of angiogenesis. It would be of interest to evaluate whether these small molecule drugs affect EVPs as opposed to anti-VEGF-A.

We agree with reviewer that other modalities to block VEGF signaling exist. However, these kinase inhibitors lack specificity and even if in a clinical context they could be useful, for the purpose of our study they affect both VEGFR2 signaling as well as PDGFR signaling. PDGFR is highly expressed on EVPs whereas VEGFR2 is highly expressed on D cells. Using these small molecule inhibitors would not allow to selectively target one population over the other.

- Several studies argued that the contribution of endothelial progenitor cells to tumor growth and metastasis is minimal if at all. This has not been discussed in the context of this study, and it should as this area of research is under heavy debate.

We agree with reviewer that the concept of endothelial progenitor is under debate. We and others have recently established a new definition of these progenitors based on strict stem cell biology criteria. These include self-renewal, plasticity and lineage tracing. We report similar findings in mice and humans. We cannot be certain that many of the previous studies on endothelial progenitors followed these functional requirements that

in our view are essential. We provide now an explanation why our results might differ from others. This has been amended in the text on page 18.

- Figure 2B, can the authors immunostain CD31 blood vessels. In other words, how many of the vessels in the tumor are YFP+ and how many of them are angiogenesis per se (not vasculogenesis)?

We thank the review for this comment. We believe that most of our work has focused on the vascularization of the center of the tumor where we observe these vasculogenic events. We have observed the tumor periphery as being populated by many vessels probably resulting from angiogenesis of surrounding vessels. We have now clarified this in the discussion.

To address specifically your request, we have now performed lineage tracing using the Sox18Cre model and have counted the proportion of labelled cells in the center versus the periphery. The Sox18Cre best reflects in earlier time-points the EVP populations both in the center and at the periphery.

As you can see in the figure below about a third of CD31+ vessels are YFP+ both in the center and the periphery at D15 post tumor inoculation (Tamoxifen was injected at D3 post tumor inoculation). This suggests that about a third of vessels are derived from the vasculogenic process described here assuming that in this model Cre efficiency is at 100%. To examine Cre efficiency in these conditions we did the same measurements in the Cdh5Cre model. In this model injected with the same quantity and timing of tamoxifen, about 40% of CD31+ vessels were YFP+ both in the center and at the periphery of tumors where in theory 100% would be expected. Overall, we are not certain the questions can be addressed easily and reliably as suggested by reviewer given that Cre efficiency does not allow staining a large proportion of Cre expressing cells.

References:

1. Patel, J. *et al.* Functional Definition of Progenitors Versus Mature Endothelial Cells Reveals Key SoxF-Dependent Differentiation Process. *Circulation* **135**, 786-805 (2017).
2. He, L. *et al.* Preexisting endothelial cells mediate cardiac neovascularization after injury. *J Clin Invest* **127**, 2968-2981 (2017).
3. Shafiee, A., Patel, J., Hutmacher, D.W., Fisk, N.M. & Khosrotehrani, K. Meso-Endothelial Bipotent Progenitors from Human Placenta Display Distinct Molecular and Cellular Identity. *Stem Cell Reports* **10**, 890-904 (2018).

REVIEWERS' COMMENTS:

Reviewer #1 (Remarks to the Author):

I am satisfied by the additional data and textual modifications in the revised manuscript in response to my questions.

Reviewer #2 (Remarks to the Author):

The authors mostly addressed my comments. I have no additional comments.